# Sampling from the latent space in Autoencoders: A simple way towards generative models?

## Abstract

By sampling from the latent space of an autoencoder and decoding the latent space samples to the original data space, any autoencoder can simply be turned into a generative model. For this to work, it is necessary to model the autoencoder's latent space with a distribution from which samples can be obtained. Several simple possibilities (kernel density estimates, Gaussian distribution) and more sophisticated ones (Gaussian mixture models, copula models, normalization flows) can be thought of and have been tried recently. This study aims to discuss, assess, and compare various techniques that can be used to capture the latent space so that an autoencoder can become a generative model while striving for simplicity. Among them, a new copula-based method, the *Empirical Beta Copula Autoencoder*, is considered. Furthermore, we provide insights into further aspects of these methods, such as targeted sampling or synthesizing new data with specific features.

## 1 Introduction

Generating realistic sample points of various data formats has been of growing interest in recent years. Thus, new algorithms such as *Autoencoders (AEs)* and *Generative Adversarial Networks (GANs)* Goodfellow et al. (2014) have emerged. GANs use a discriminant model, penalizing the creation of unrealistic data from a generator and learning from this feedback. On the other hand, AEs try to find a low-dimensional representation of the high-dimensional input data and reconstruct from it the original data. To turn an AE into a generative model, the low-dimensional distribution is modeled, samples are drawn, and thereupon new data points in the original space are constructed with the decoder. We call this low dimensional representation of the data in the autoencoder the *latent space* in the following. Based on that, *Variational Autoencoders (VAEs)* have evolved, optimizing for a Gaussian distribution in the latent space Kingma & Welling (2014). Adversarial autoencoders (AAEs) utilize elements of both types of generative models, where a discriminant model penalizes the distance of the encoded data from a prior (Gaussian) distribution (Makhzani et al., 2016). However, such strong (and simplifying) distributional assumptions as in the VAE or AAE can have a negative impact on performance, leading to a rich literature coping with the challenge of reducing the gap between approximate and true posterior distributions (e.g., Rezende & Mohamed 2015; Tomczak & Welling 2018; Kingma et al. 2016; Gregor et al. 2015; Cremer et al. 2018; Marino et al. 2018; Takahashi et al. 2019). In this paper we discuss more flexible approaches modeling the latent space without imposing restrictions on the underlying distribution.

Recently, Tagasovska et al. 2019 presented the *Vine Copula Autoencoder (VCAE)*. Their approach comprises two building blocks, an autoencoder and a vine copula which models the dependence structure in latent space. By that, they were able to create realistic, new images with samples from the fitted vine copula model in the latent space. In this work, we want to elaborate on this idea and compare various methods to model the latent space of an autoencoder to turn it into a generative model. To this end, we analyze, amongst others, the usage of *Gaussian mixture models (GMM)* as done by Ghosh et al. 2020, the vine copula approach by Tagasovska et al. 2019, and simple multivariate *Kernel Density Estimates*. Additionally, we introduce a new, non-parametric copula approach, the *Empirical Beta Copula Autoencoder (EBCAE)*. To get a deeper understanding of how this can turn a standard autoencoder into a generative model, we inspect resulting images, check the models for their ability to generalize and compare additional features. In this

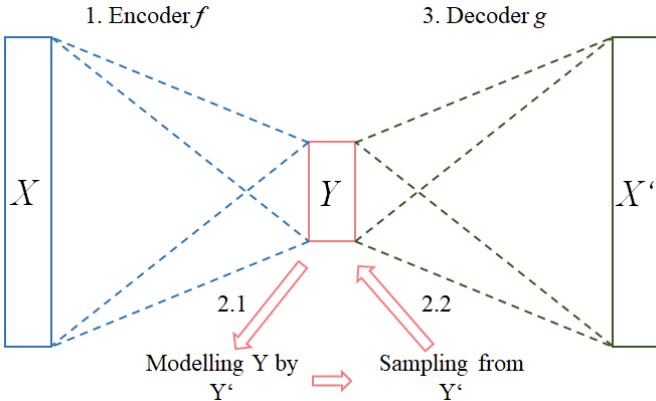

Figure 1: Function scheme of simple generative autoencoders. 1. An encoder $f$ encodes the data $X$ to a low dimensional representation $Y$. 2.1 $Y$ is modeled by $Y'$, 2.2 Generate new synthetic samples of the latent space by sampling from $Y'$. 3. Decode the new samples with the decoder $g$.

study we do not aim to beat the latest SOTA generative models but want to shed light on different modeling techniques in the latent space and their characteristics in a rather straightforward autoencoder setting, which may be applied in more sophisticated models as well. Thus, we strive for simplicity and take an alternative route to more and more complex models. We believe that such an analysis in a straightforward setting is essential for understanding the effects from different sampling methods, which may then be applied in more advanced generative models. We also check whether the methods may be a simple alternative to more complex models, such as normalization flows Rezende & Mohamed (2015) or diffusion models (see, e.g., Rombach et al. 2022; Vahdat et al. 2021). More specifically, we use the well-known Real NVP (Dinh et al., 2017) as an example from these more sophisticated machine learning models in the latent space but do not elaborate on these in detail. Note that in contrast to other methods (e.g., as proposed by Oring et al. 2021, Berthelot et al. 2019 or van den Oord et al. 2017), the investigated overall approach does not restrict or change the training of the autoencoder in any form. All models considered in this work are constructed in three steps, visualized in Figure 1. First, an autoencoder, consisting of an encoder $f$ and a decoder $g$, is trained to find a low-dimensional representation of the data $X$. Second, the data in the latent space $Y$ is used to learn the best fitting representation $Y'$ of it. This is where the examined models differ from each other by using different methods to model the latent space. Finally, we sample from the learned representation of the latent space and feed the samples into the decoder part of the autoencoder, creating new synthetic data samples.

Generative models are a vivid part of the machine learning literature. For example, new GAN developments Varshney et al. (2021); Karras et al. (2021); Lee et al. (2021); Hudson & Zitnick (2021), developments in the field of autoencoders, Larsen et al. (2016); Yoon et al. (2021); Zhang et al. (2020); Shen et al. (2020) or developments in variational autoencoders Sohn et al. (2015); Havtorn et al. (2021); Masrani et al. (2019); Xu et al. (2019) are emerging. We again want to emphasize that for the models we consider, no prior is needed, nor the optimization approach is changed, i.e., the latent space is modeled after the training of the autoencoder post-hoc. Thus, the presented approach could be transferred to other, more sophisticated, state-of-the-art autoencoders, as hinted in Ghosh et al. 2020. The general idea of creating new data by sampling in the latent space of a generative model has already been used by, e.g., Tagasovska et al. 2019; Dai & Wipf 2019; Brehmer & Cranmer 2020 or Ghosh et al. 2020, but to the best of our knowledge, no analysis and comparison of such methods have been made so far. Closely related, more and more researchers specifically address the latent space of generative models Mishne et al. (2019); Fajtl et al. (2020); Moor et al. (2020); Oring et al. (2021); Hofert et al. (2021) in their work. There, especially hierarchical methods as suggested by Maaløe et al. (2019) seem to be promising. Further, Autoencoders based on the Wasserstein Distance lately achieved excellent results by changing the regularization term of a VAE and using or learning a Gaussian Mixture Prior Tolstikhin et al. (2019); Mondal et al. (2021), analogously to our use of Gaussian Mixtures fitting the latent space distribution.

This work does not propose a new 'black-box algorithm' for generating data (although we present the new EBCAE) but analyses challenges and possible answers on how autoencoders can be turned into generative models by using well-understood tools of data modeling. One of our main findings is, that is hard to find a trade-off between out-of-bound sampling and creating new pictures. We conclude that besides a pure numerical perspective and looking at new random samples of a generative model with a latent space, the resulting image of the nearest neighbor in the latent space from the training data should be inspected. We demonstrate in our experiments that copula-based approaches may be promising alternatives to traditional modeling methods since they allow for the recombination of marginal distributions from one class with the dependence structure of another class leading to new possibilities in synthesizing images and discuss targeted sampling. Our conclusion is intended to point out relevant aspects to the user and discusses the advantages and disadvantages of the models examined.

The remainder of the paper is structured as follows. Section 2 introduces various methods for modeling the latent space. Besides traditional approaches, copula-based methods are introduced. Section 3 describes the implementation, evaluation, and results of the experiments carried out. In Section 4 we discuss the results and conclude the paper. Last, we provide additional experiments and insides for interested readers in the appendix.

## 2 Modeling the latent space

In this section, we want to introduce and reflect on different methods to model the latent space in an autoencoder (Step 2 in Figure 1). All methods aim to fit the low-dimensional data $Y$ as best as possible to be able to create new sample points in the latent space, which leads to new realistic images after passing the decoder. We first recap more 'traditional' statistical tools, followed by copulas as an intuitive and flexible tool for modeling high-dimensional data. We briefly explain how each approach can be used to model data in the latent space and how to obtain samples thereof. Note that we do not introduce our benchmark models, namely the standard plain vanilla *VAE* and the *Real NVP*, and refer to the original papers instead (Kingma & Welling, 2014; Dinh et al., 2017). Pseudocode of the overall sampling approach is given in the Appendix (Algorithm 2).

### 2.1 Traditional modeling methods

We classify the *multivariate Gaussian distribution*, a *Kernel Density Estimation (KDE)*, and a *Gaussian Mixture Model (GMM)* as traditional modeling methods and give a rather short treatment of each below. They are well known and can be studied in various statistics textbooks such as Hastie et al. 2001 or Bishop 2006.

### Multivariate Gaussian

The probably simplest method is to assume the data in the latent space to follow a multivariate Gaussian distribution. Thus, we estimate the covariance matrix $\hat{\Sigma}$ and mean vector $\hat{\mu}$ of the date $Y$. In the second step, we draw samples thereof and pass them through the decoder to generate new images. Note that this is similar to the sampling procedure in a VAE, but without forcing the latent space to be Gaussian during training.

### GMM

The *Gaussian Mixture Model (GMM)* aims to model the density of the latent space by mixing $M$ multivariate Gaussian distributions. Thus, the Gaussian mixture model has the form

$$f(x) = \sum_{m=1}^{M} \alpha_m \phi(x; \mu_m, \Sigma_m) \tag{1}$$

where $\alpha_m$ denotes the mixing parameter and $\phi$ the density of the multivariate normal distribution with mean vector $\mu_m$ and covariance matrix $\Sigma_m$. The model is usually fit by maximum likelihood using the EM algorithm. By combining several Gaussian distributions, it is more flexible than estimating only one Gaussian

distribution as above. A GMM can be seen as some kind of kernel method (Hastie et al., 2001), having a rather wide kernel. In the extreme case, i.e., where $m$ equals the number of points the density is estimated on, a Gaussian distribution with zero variance is centered over each point. Kernel density estimation is introduced in the following.

**KDE**

*Kernel Density Estimation* is a well-known non-parametric tool for density estimation. Put simply, a KDE places a density around each data point. The total resulting estimated density is constructed by

$$f(x) = \frac{1}{N\lambda} \sum_{i=1}^{N} K_\lambda(x_0, x_i) \tag{2}$$

with $N$ being the total number of data points, $\lambda$ the bandwidth, and $K$ the used kernel. Note that the choice of bandwidth and kernel can affect the resulting estimated density. The kernel density estimation can be performed in univariate data as well as in multivariate data. In this work, we rely on the most commonly used kernel, the Gaussian Kernel, and a bandwidth fitted via *Silverman's rule of thumb* (Silverman, 1986) for the univariate KDEs (i.e. for estimating the marginal distributions of the latent space), while we use a grid search with 10-fold cross-validation in the multivariate case.

We use kernel density estimation in multiple manners throughout this work. First, we use a multivariate KDE to model the density of the data in the latent space itself. In the case of a Gaussian kernel, it can be written by

$$f(x) = \frac{1}{N\sqrt{\Sigma}2\pi} \sum_{i=1}^{N} e^{-1/2(x-x_i)'\Sigma^{-1}(x-x_i)} \tag{3}$$

where $\Sigma$ represents the covariance matrix of the kernel, i.e., the matrix of bandwidths. Second, we ignore the dependence structure between margins and estimate the univariate densities of each dimension in the latent space by a KDE for each marginal distribution. In this way, we are able to find out whether explicitly modeling the dependence structure is necessary or not. We call that approach the *Independent modeling approach* also denoted short by *Independent* in the following. Last, we use univariate KDEs for modeling the marginal distributions of each dimension in the latent space and use them in the copula models described below.

## 2.2 Copula based models

Besides the traditional modeling methods introduced above, we apply copula based models. In the following, we first introduce copulas as a tool for high-dimensional data, which allows us to model the latent space in our application. Then, we focus on the two copula-based methods to model the latent space of the autoencoder: the *vine copula* and the *empirical beta copula* approach. For detailed introductions to copulas, we refer the reader to Nelsen 2006; Joe 2014; Durante & Sempi 2015.

*Copulas* have been subject to an increasing interest in the *Machine Learning* community over the last decades, see, e.g., Dimitriev & Zhou 2021; Janke et al. 2021; Messoudi et al. 2021; Ma et al. 2021; Letizia & Tonello 2020; Liu 2019; Kulkarni et al. 2018; Tran et al. 2015. In a nutshell, copula theory enables us to decompose any $d$-variate distribution function into $d$ marginal univariate distributions and their joint dependence structure, given by the copula function. Thus, copulas "couple" multiple univariate distributions into one joint multivariate distribution. More formally, a $d$-variate copula $C : [0,1]^d \rightarrow [0,1]$ is a $d$-dimensional joint distribution function whose margins are uniformly distributed on the unit interval. Decomposing and coupling distributions with copulas is formalized in Theorem 2.1 going back to Sklar 1959.

**Theorem 2.1** (Sklar 1959)**.** *Consider a d-dimensional vector of random variables $\boldsymbol{Y_i} = (Y_{i,1}, \ldots, Y_{i,d})$ with joint distribution function $F_{\boldsymbol{Y}}(y_i) = P(Y_1 \leq y_{i,1}, \ldots, Y_d \leq y_{i,d})$ for $i = 1, \ldots, n$. The marginal distribution*

functions $F_j$ are defined by $F_j(y_{i,j}) = P(Y_j \leq y_{i,j})$ for $y_{i,j} \in \mathbb{R}$, $i = 1, \ldots, n$ and $j = 1, \ldots, d$. Then, there exists a copula $C$, such that

$$F_{\boldsymbol{Y}}(y_1, .., y_d) = C(F_1(y_1), \ldots, F_d(y_d))$$

for $(y_1, \ldots, y_d) \in \mathbb{R}^d$. Vice versa, using any copula $\tilde{C}$, it follows that $\tilde{F}_{\boldsymbol{Y}}(y_1, .., y_d) := \tilde{C}(F_1(y_1), \ldots, F_d(y_d))$ is a proper multivariate distribution function.

This allows us to construct multivariate distributions with the same dependence structure but different margins or multivariate distributions with the same margins but different couplings/pairings, i.e., dependence structures. The simplest estimator is given by the empirical copula. It can be estimated directly on the ranks of each marginal distribution by

$$\hat{C}(\mathbf{u}) = \frac{1}{n} \sum_{i=1}^{n} \prod_{j=1}^{d} \mathbf{1} \left\{ \frac{r_{i,j}^{(n)}}{n} \leq u_j \right\} \tag{4}$$

with $\mathbf{u} = (u_1, \ldots, u_d) \in [0,1]^d$ and $r_{i,j}^{(n)}$ denoting the rank of each $y_{i,j}$ within $(y_{1,j}, \ldots, y_{n,j})$, i.e.,

$$r_{i,j}^{(n)} = \sum_{k=1}^{n} \mathbf{1}\{y_{k,j} \leq y_{i,j}\}. \tag{5}$$

Note that $\mathbf{u} = (u_1, \ldots, u_d)$ represents a quantile level, hence a scaled rank. Simultaneously, the univariate margins can be estimated using a KDE so that the full distribution latent space is governed for. Note that it is not possible to draw new samples from the empirical copula directly as no random process is involved. In our applications, the latent space is typically equipped with dimensions $\geq 2$. Although a variety of two-dimensional copula models exist, the amount of multivariate (parametric) copula models is somewhat limited. We present two solutions to this problem in the following, namely *vine copulas* and the *empirical beta copula*.

**Vine Copula Autoencoder**

*Vine copulas* decompose the multivariate density as a cascade of bivariate building blocks organized in a hierarchical structure. This decomposition is not unique, and it influences the estimation procedure of the model. Here, we use *regular-vine (r-vine)* models Czado (2019); Joe (2014) to model the 10, 20 and 100 dimensional latent space of the autoencoders at hand. An r-vine is built of a sequence of linked trees $T_i = (V_i, E_i)$, with nodes $V_i$ and edges $E_i$ for $i = 1, \ldots, d-1$ and follows distinct construction rules which we present in Appendix B.

The $d$-dimensional copula density can then be written as the product of its bivariate building blocks:

$$c(u_1, \ldots, u_d) = \prod_{i=1}^{d-1} \prod_{e \in E_i} c_{a_e b_e; D_e}(u_{a_e | D_e}, u_{b_e | D_e}) \tag{6}$$

with conditioning set $D_e$ and conditional probabilities, e.g., $u_{a_e | D_e} = \mathbb{P}(U_{a_e} \leq u_{a_e} | D_e)$. The conditioning set $D_e$ includes all variables conditioned on at the respective position in the vine structure (see Appendix B). For each resulting two-dimensional copula of conditional variables, any parametric or non-parametric copula model (as done by Tagasovska et al. 2019) can be chosen. However, the construction and estimation of vine copulas is rather complicated. Hence, assuming independence for seemingly unimportant building blocks, so-called truncation, is regularly applied. Because of this, truncated vine copula models do not capture the complete dependence structure of the data, and their usage is not underpinned by asymptotic theory. We refer to Czado (2019); Czado & Nagler (2022); Aas (2016) for reviews of vine copula models.

**Empirical Beta Copula Autoencoder**

The *empirical beta copula* (Segers et al., 2017) avoids the problem of choosing a single, parametric multivariate copula model due to its non-parametric nature. Further, and in contrast to the presented vine copula

approach, it offers an easy way to model the full, non-truncated multivariate distribution based on the univariate ranks of the joint distribution and, thus, seems to be a reasonable choice to model the latent space. The empirical beta copula is closely related to the empirical copula (see Formula 5) and is a crucial element of the Empirical-Beta-Copula Autoencoder. It is solely based on the ranks $r_{i,j}^{(n)}$ of the original data $\mathbf{Y}$ and can be interpreted as a continuous counterpart of the empirical copula. It is defined by

$$C^\beta = \frac{1}{n} \sum_{i=1}^{n} \prod_{j=1}^{d} F_{n,r_{i,j}^{(n)}}(u_j) \tag{7}$$

for $\mathbf{u} = (u_1, \ldots, u_d) \in [0,1]^d$, where

$$F_{n,r_{i,j}^{(n)}}(u_j) = P(U_{(r_{i,j}^{(n)})} \leq u_j) \tag{8}$$

$$= \sum_{p=r_{i,j}^{(n)}}^{n} \binom{n}{p} u_j^p (1 - u_j)^{(n-p)} \tag{9}$$

is the cumulative distribution function of a *beta distribution*, i.e., $\mathbb{B}(r_{i,j}^{(n)}, r_{i,j}^{(n)} + n - 1)$. As $r_{i,j}$ is the rank of the $i^{\text{th}}$ element in dimension $j$, $U_{(r_{i,j})}$ represents the $r_{i,j}{}^{\text{th}}$ order statistic of $n$ i.i.d. uniformly distributed random variables on $[0,1]$. For example, if the rank of the $i^{\text{th}}$ element in dimension $j$ is 5, $U_{(r_{i,j})} = U_{(5)}$ denotes the $5^{\text{th}}$ order statistic on $n$ i.i.d. uniformly distributed random variables.

The intuition behind the empirical beta copula is as follows: Recall that the marginal distributions of a copula are uniformly distributed on $[0,1]$ and, hence, the $k^{\text{th}}$ smallest value of scaled ranks $r_{i,j}^{(n)}/n$ corresponds to the $k^{\text{th}}$ order statistic $U_{(k)}$. Such order statistics are known to follow a *beta distribution* $\mathbb{B}(k, k+n-1)$ (David & Nagaraja, 2003). Consequently, the mathematical idea of the empirical beta copula is to replace each indicator function of the empirical copula with the cumulative distribution function of the corresponding rank $r_{i,j}^{(n)}$.

We argue that the empirical beta copula can be seen as the naturally extended version of the empirical copula, thus, it seems to be a good choice for dependence modeling. Segers et al. 2017 further demonstrates that the empirical beta copula outperforms the empirical copula both in terms of bias and variance. A theorem stating the asymptotic behavior of the empirical copula is given in Appendix C.

Synthetic samples in the latent space $y'$ are created by reversing the modeling path. First, random samples from the copula model $\mathbf{u} = (u_1, \ldots, u_d)$ are drawn. Then, the copula samples are transformed back to the natural scale of the data by the inverse probability integral transform of the marginal distributions, i.e., $y'_j = \hat{F}_j(u_j)$, where $\hat{F}_j$ is the estimated marginal distribution and $u_j$ the $j$th element of the copula sample for $j \in \{1, \ldots, d\}$. Algorithm 1 summarizes the procedure.

---

**Algorithm 1:** Sampling from Empirical Beta Copula

---

**Input:** Sample $Y \subset \mathbb{R}^{n \times d}$, new sample size $m$
**begin**

    Compute rank matrix $R^{n \times d}$ out of $Y$
    Estimate marginals of $Y$ with KDE, $\hat{f}_1(y_1), \ldots, \hat{f}_d(y_d)$.
    **for** $i \leq m$ **do**
        Draw random from $I \in [1, \ldots, n]$
        **for** $j \leq d$ **do**
            Draw $u_{I,j} \sim \mathbb{B}(R_{Ij}, n + 1 - R_{Ij})$
        Set $u_i = (u_{I1}, \ldots, u_{Id})$
        Rescale margins by $Y_i = \hat{F}_1^{-1}(u_{i1}), \ldots, \hat{F}_d^{-1}(u_{id})$.

**Output:** New sample $Y'$ of size m

---

We now present the experiments and results of a comparative study including all mentioned methodologies to model the latent space in the next section.

## 3 Experiments

In this section, we present the results of our experiments. We use the same architecture for the autoencoder in all experiments for one dataset but replace the modeling technique for the latent space for all algorithms. The architecture, as well as implementation details, are given in Appendix D. We further include a standard VAE and the Real NVP normalization flow approach modeling the latent space in our experiments to serve as a benchmark.

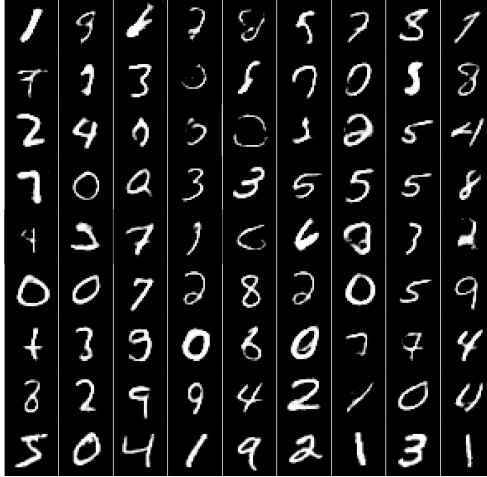 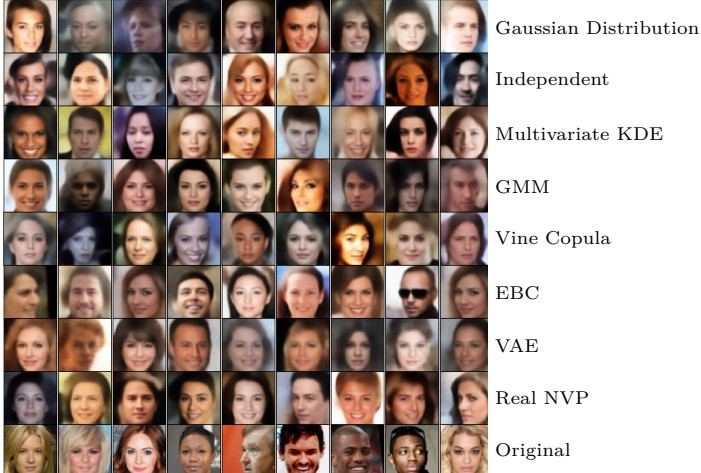

Gaussian Distribution

Independent

Multivariate KDE

GMM

Vine Copula

EBC

VAE

Real NVP

Original

Figure 2: Comparison of random, synthetic samples of different Autoencoder models row by row for MNIST (left) and CelebA (right). Original input samples are given in the last row.

### 3.1 Setup

We first describe the overall methodology and the usage of the methods proposed in Section 2. We then introduce the used data sets and evaluation framework.

**Methodology**

We train an autoencoder consisting of two neural nets, an *encoder $f$*, and a *decoder $g$*. The encoder $f$ maps data $X$ from the original space to a lower-dimensional space, while the decoder $g$ reconstructs this low-dimensional data $Y$ from the low-dimensional latent space to the original space (see Fig. 1). We train both neural nets in a way that the reconstruction loss is minimized, i.e., that the reconstructed data $X' = g(f(X))$ is as similar to the original data $X$ as possible. In the second step, we model the latent space $Y$ data with a multivariate Gaussian distribution, a Gaussian mixture model, Kernel density estimates, the two presented copula methods and the Real NVP. Thus, we fit models with different flexibility and complexity while keeping the training process of the autoencoder untouched. Last, new samples are generated by decoding random samples from the learned model in the latent space. Note that such an approach is only reasonable when the underlying autoencoder has learned a relevant and interesting representation of the data and the latent space is smooth. We demonstrate this in Appendix E.

**Datasets**

We conduct experiments on one small-scale, one medium, and one large-scale dataset. The small-scale *MNIST* dataset (LeCun et al., 2010) includes binary images of digits, while the medium-scale *SVHN* dataset (Netzer et al., 2011) contains images of house numbers in Google Street View pictures. The large-scale *CelebA* dataset (Liu et al., 2015) consists of celebrity images covering 40 different face attributes. We split data into a train set and a test set of 2000 samples which is a commonly used size for evaluation (Tagasovska et al., 2019;

Xu et al., 2018). Note that the data sets cover different dimensionalities in the latent space, allowing for a throughout assessment of the methods under investigation.

**Evaluation**

Evaluation of results is performed in several ways. First, we visually compare random pictures generated by the models. Second, we evaluate the results with the framework proposed by Xu et al. 2018, since a log-likelihood evaluation is known to be incapable of assessing the quality (Theis et al., 2016) and unsuitable for non-parametric models. Based on their results, we choose five metrics in our experiments: The *earth mover distance (EMD)*, also known as *Wasserstein distance* (Vallender, 1974); the *mean maximum discrepancy (MMD)* (Gretton et al., 2007); the *1-nearest neighbor-based two-sample test (1NN)*, a special case of the classifier two-sample test (Lopez-Paz & Oquab, 2017); the *Inception Score* (Salimans et al., 2016); and the *Fréchet inception distance* (Heusel et al., 2017) (the latter two over ResNet-34 softmax probabilities). In line with Tagasovska et al. 2019 and as proposed by Xu et al. 2018, we further apply the EMD, MMD, and 1NN over feature mappings in the convolution space over ResNet-34 features. For all metrics except the Inception Score, lower values are preferred. For more details on the metrics, we refer to Xu et al. 2018. Next, we evaluate the ability to generate new, realistic pictures by the different latent space modeling techniques. Therefore, we compare new samples with their nearest neighbor in the latent space stemming from the original data. This shows us whether the learned distribution covers the whole latent space, or stays too close to known examples, i.e., the model does not generalize enough. Finally, we compare other features of the tested models, such as their ability of targeted sampling and of recombining attributes.

## 3.2 Results

In the following, we show results for our various experiments. First, we present visual results for each of the methods investigated to gain a qualitative understanding of their differences. Second, we compare the methods in terms of performance metrics. Third, we evaluate the latent space and nearest neighbots in the latent space. Finally, we address computing times and discuss targeted sampling and recombination of image features.

**Visual Results**

Figure 2 shows images generated from each method for MNIST and CelebA. The GMM model is composed of 10 elements, and the KDE is constructed using a Gaussian kernel with a bandwidth fitted via a grid search and 10-fold cross-validation. The specification of the Real NVPs are given in the Appendix.

For the MNIST dataset, we observe the best results for the EBCAE (row 6) and KDE (row 3), while the other methods seem to struggle a bit. For the CelebA, our visual observations are slightly different. All methods produce images that are clearly recognizable as faces. However, the Gaussian samples in row 1 and independent margins in row 2 create pictures with some unrealistic artefacts, blurry backgrounds, or odd colors. This is also the case for the GMM in row 4 and VCAE in row 5, but less severe. We believe that this comes from samples of an empty area in the latent space, i.e., where none of the original input pictures were projected to. In contrast to that, the samples in the latent space of the KDE, EBCAE, and Real NVP stay within these natural bounds, producing good results after passing the decoder (rows 3, 6, 8). Recall that all methods use the same autoencoder and only differ by means of sampling in the latent space. From our observations, we also conclude that the autoencoder for the CelebA dataset is less sensitive toward modeling errors in the latent space since all pictures are clearly recognizable as faces. In contrast, for the MNIST dataset, not all images clearly show numbers. Similar results for SVHN are presented in the Appendix.

**Numerical Results**

The numerical results computed from 2000 random samples displayed in Figure 3 prove that dependence truly matters within the latent space. Simultaneously, the KDE, GMM, and EBCAE perform consistently well over all metrics, delivering comparable results to the more complex Real NVP. Especially the EBCAE outperforms the other methods, whereas the VCAE, Gauss model, and VAE usually cluster in the middle.

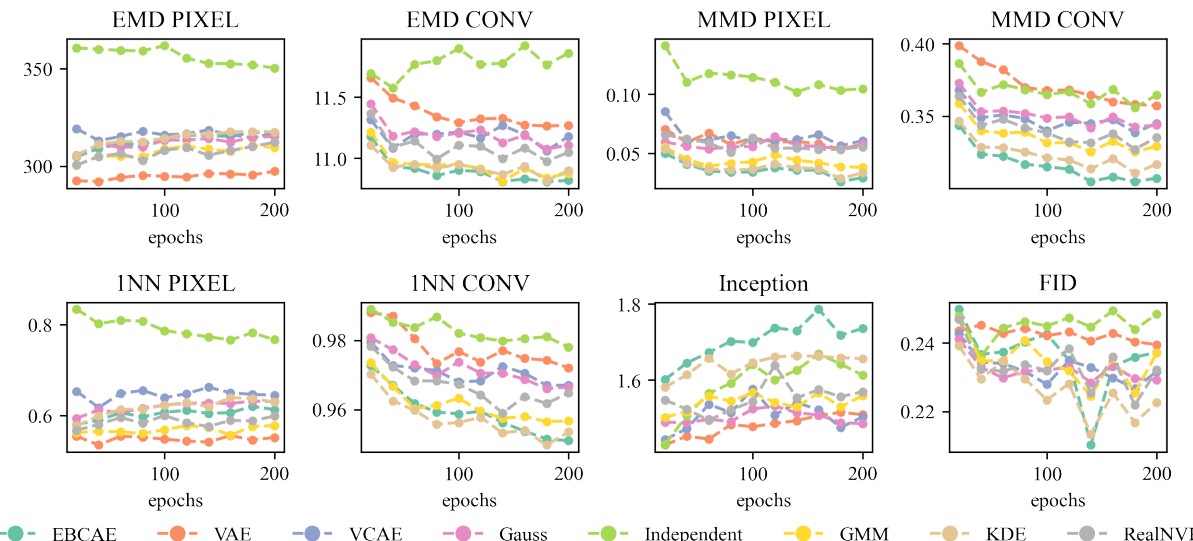

Figure 3: Performance metrics of generative models on **CelebA**, reported over epochs computed from 2000 random samples. Note that they only differ in the latent space sampling and share the same autoencoder.

We further report results over the number of samples in the latent space in Figure 9 in the Appendix. This, at first sight, unusual perspective visualizes the capability to reach good performance even for small sample sizes in latent space. In a small-sample regime, it is crucial to assess how fast a method adapts to data in the latent space and models it correctly. We see that all methods perform well for small sample sizes, i.e., $n = 200$. Similar experiments for MNIST and SVHN can be found in Appendix F.

**Nearest Neighbour and Latent Space Evaluation**

Next, we evaluate the different modeling techniques in their ability to generate new, realistic images. For this, we focus on pictures from the CelebA dataset in Figure 4. First, we create new, random samples with the respective method (top row) and then compare these with their decoded nearest neighbor in the latent space (middle row). The bottom row displays the latent space nearest neighbor in the original data space before applying the autoencoder. By doing so, we are able to disentangle two effects. First, the effect from purely encoding-decoding an image and, second, the effect of modeling the latent space. Thus, we can check whether new images are significantly different from the input, i.e., whether the distribution modeling the latent space merely reproduces images or generalizes to some extent.

We observe that the samples from GMM, VCAE and the Real NVP substantially differ from their nearest neighbors. However, again they sometimes exhibit unrealistic colors and blurry backgrounds. The samples created from KDE and EBCAE look much more similar to their nearest neighbors in the latent space, indicating that these methods do not generalize to the extent of the other methods. However, their samples do not include unrealistic colors or features and seem to avoid sampling from areas where no data point of the original data is present. Thus, they stay in 'natural bounds'. Note that this effect apparently is not reflected in the numerical evaluation metrics. We, therefore, recommend that, in addition to a quantitative evaluation, a qualitative evaluation of the resulting images should always be performed.

To further underpin this point, Figure 5 shows 2-dimensional TSNE-Embeddings (see, e.g.,van der Maaten & Hinton 2008) of the latent space for all six versions of the autoencoder (MNIST). Black points indicate original input data, and colored points are synthetic samples from the corresponding method. We see that the KDE, as well as the EBCAE, stay close to the original space. The samples from the GMM and Real NVP also seem to closely mimic the original data, whereas the other methods fail to do so. This visualization confirms our previous conjecture that some algorithms tend to sample from 'empty' areas in the latent space, leading to unrealistic results.

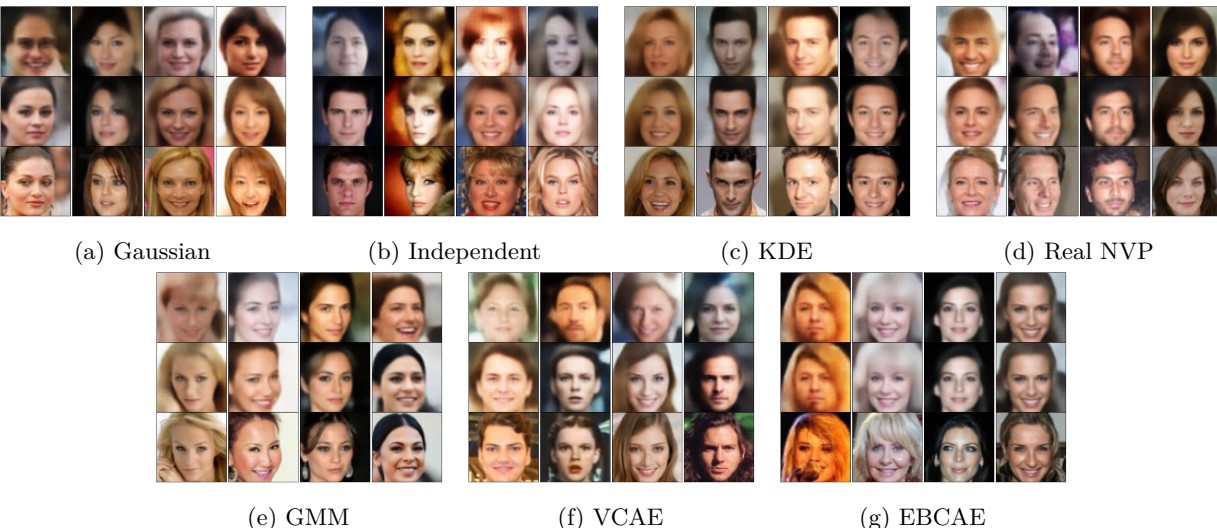

(a) Gaussian  (b) Independent  (c) KDE  (d) Real NVP

(e) GMM  (f) VCAE  (g) EBCAE

Figure 4: Nearest neighbor evaluation of the six investigated modeling methods after decoding. **Top row:** Newly generated images. **Middle row:** Nearest neighbor of new image in the latent space of training samples after decoding. **Bottom row:** Original input training picture of nearest neighbor in latent space.

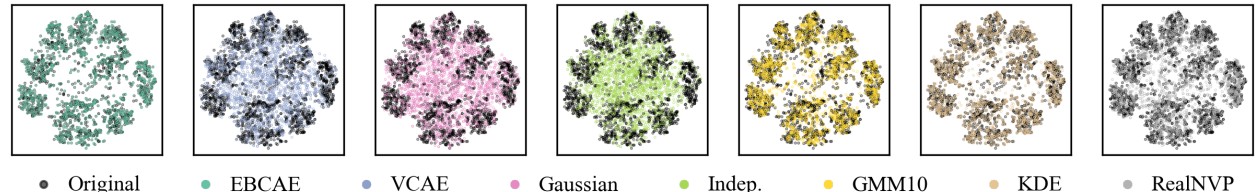

● Original  ● EBCAE  ● VCAE  ● Gaussian  ● Indep.  ● GMM10  ● KDE  ● RealNVP

Figure 5: TSNE embeddings of samples in the latent space of the **MNIST** dataset. Points from the original input training data $Y$ are given in black, whereas new, synthetic samples $Y'$ in the latent space stemming from the different modeling methods are colored.

**Computing Times, Targeted Sampling and Recombination**

We also report computing times for learning and sampling of the different models for MNIST and CelebA in Table 1. Unsurprisingly, the more straightforward methods such as Gauss, Independence, KDE, and GMM, exhibit the lowest sampling times. The Real NVP shows the highest learning time as a neural network is fitted. However, we expect the difference to be much smaller once trained on an appropriate GPU. The times also reflect the complexities of the methods in the latent space dimensions.

Table 1: Modeling and sampling time in the **CelebA** and **MNIST** dataset of 2000 artificial samples based on a latent space of size $n = 2000$ in [s].

| Method | CelebA Learn | CelebA Sample | MNIST Learn | MNIST Sample |
|---|---|---|---|---|
| Gauss | <0.01 | 0.01 | 0.002 | 0.002 |
| Indep. | 4.10 | 0.07 | 0.393 | 0.003 |
| KDE | 75.25 | 0.01 | 13.958 | 0.001 |
| GMM | 1.35 | 0.03 | 0.115 | 0.004 |
| VCAE | 306.97 | 148.48 | 10.345 | 4.590 |
| EBCAE | 3.41 | 59.36 | 0.328 | 5.738 |
| Real NVP | 2541.19 | 3.69 | 341.608 | 0.477 |

Last, we discuss other features of the tested methods, such as targeted sampling and recombination. In contrast to the other techniques, the KDE and EBCAE allow for targeted sampling. Thus, we can generate new images with any desired characteristic directly, e.g., only ones in a data set of images of numbers. In the case of the KDE, this simply works by sampling from the estimated density of the corresponding sub-group. In the case of the EBCAE, we randomly choose among rows in the rank matrix of original samples that share the desired specific attribute, i.e., we sample $I$ in the first for-loop in Algorithm 1 conditional on the sub-group. Thus, newly generated samples stay close to the original input and therefore share the same main characteristics. Other approaches are also possible, however, they need further tweaks to the model, training, or sampling as the *conditional variational autoencoder* (Sohn et al., 2015).

The second feature we discuss is recombination. By using copula-based models (VCAE and EBCAE), we can facilitate the decomposition idea and split the latent space in its dependence structure and margins, i.e., we combine the dependence structure of images with a specific attribute with the marginal distributions of images with different attributes. Therefore, copula-based methods allow controlling the attributes of created samples to some extent. Our experiments suggest that the dependence structure provides the basic properties of an image, while the marginal distributions are responsible for details (see, e.g., Figure 6). However, we want to point out that it is not generally clear what information is embedded in the dependence structure and what information is in the marginal distributions of latent space. This might also depend on the autoencoder and the dataset at hand. Thad said, using such a decomposition enables higher flexibility and hopefully fuels new methodological developments in this field.

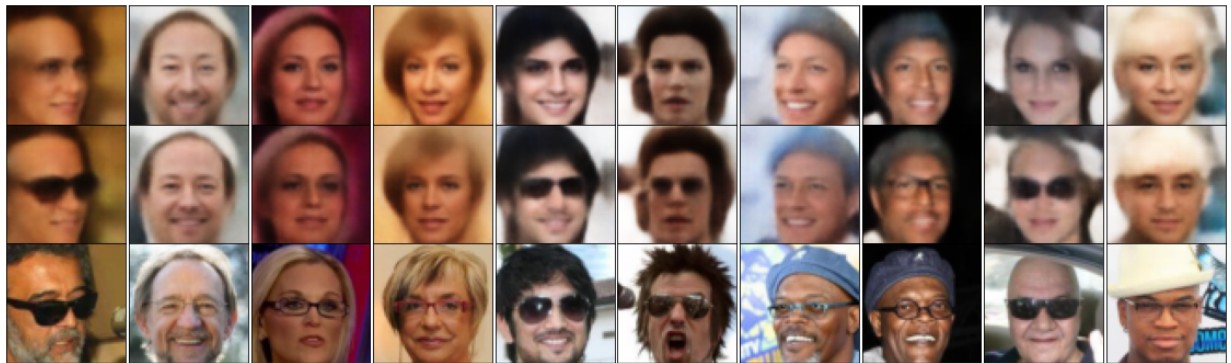

Figure 6: Samples from recombination experiment with the EBCAE. Glasses are removed by using the marginal distribution of the training data without glasses in the latent space. **Top row:** Samples created with the dependence structure in latent space from samples with glasses and marginal distributions in latent space from samples without glasses. **Middle row:** Nearest neighbor of newly created sample in the training data after decoding. **Bottom row:** Original input picture of nearest neighbor in latent space.

## 4    Discussion

In this section, we want to discuss the results of our experiments and want to express some further thoughts. So, is sampling from the latent space a simple way towards generative models? We observed that sampling from the latent space via the investigated methods is indeed a viable approach to turn an autoencoder into a generative model and may be promising for application in more advanced autoencoders. However, each modeling approach in this setting comes with its own restrictions, advantages, and problems.

We witness a trade-off between the ability to generalize, i.e., to create genuinely new pictures, and sample quality, i.e., to avoid unrealistic colors or artefacts. In cases where new data points are sampled in the neighborhood to existing points (as in the KDE or EBCAE), the newly generated data stays in somehow natural bounds and provides realistic, but not completely new, decoded samples. On the other hand, modeling the latent space too generically leads to bad-quality images. We believe this is similar to leaving the feasible set of an optimization problem or sampling from a wrong prior. While being close to actual points of the original latent space, new samples stay within the feasible set. By moving away from these points, the risk

of sampling from an unfeasible region and thus creating unrealistic new samples increases. Recombination via a copula-based approach of marginal distributions and dependence structures offers the possibility to detect new feasible regions in the latent space for the creation of realistic images. Also, interpolating by building convex combinations of two points in the latent space seems reasonable. However, without further restrictions during training (see, e.g., discussion in Ghosh et al. 2020), we cannot principally guarantee proper interpolation results. Further, we observe that the mentioned trade-off is not reflected by the performance metrics. Therefore, we strongly recommend not only checking quantitative results but also finding and analyzing the nearest neighbor in the original data to detect the pure reproduction of pictures. This also reveals that the development of further evaluation metrics could be beneficial.

A closely related issue is the choice of a parametric vs. a non-parametric modeling method in the latent space. Parametric methods can place probability mass in the latent space, where no data point of the original input data was observed. Thus, parametric methods are able to generate (truly) new data, subject to their assumption. However, if the parametric assumption is wrong, the model creates samples from 'forbidden' areas in the latent space leading to unrealistic images. In spite of this, carefully chosen parametric models can be beneficial, and even a log-likelihood is computable and traceable (although we do not use it for training). Non-parametric methods avoid this human decision and possible source of error completely but are closely bound to the empirical distribution of the given input data. Consequently, such methods can miss important areas of the latent space but create more realistic images. Furthermore, adjusting parameters of the non-parametric models, such as increasing bandwidths or lowering truncation levels, offer possibilities to slowly overcome these limitations.

Besides the major points above, the EBCAE and KDE offer an easy way of targeted sampling without additional training effort. This can be beneficial for various applications and is not as straightforward with other methods. Lastly, the investigated methods differ in their runtime. While vine copula learning and sampling is very time-intensive for high dimensions, the EBCAE is much faster but still outperformed by the competitors. For the non-copula methods, the GMM is really fast in both datasets while still capturing the dependence structure to some extent. In contrast to that, the Real NVP needs more time for training but is rather quick in generating new samples.

To sum up, we can confirm that there are indeed simple methods to turn a plain autoencoder into a generative model, which may then also be beneficial in more complex generative models. We conclude that the optimal method to do so depends on the goals of the user. Besides runtime considerations, the specific application of the autoencoder matters. For example, if one is interested in targeted sampling, EBCAE or KDE should be applied. Recombination experiments call for a copula-based approach, whereas in all cases, the trade-off between generalization and out-of-bound sampling should be considered.

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

# Appendix

## A   Pseudocode: Overall sampling approach

---
**Algorithm 2:** Overall sampling approach

---
**Input:** Autoencoder with Encoder $f$ and Decoder $g$
**begin**
    Compute latent space $Y$ by passing training samples through encoder $f$
    **for** *each method* **do**
        Model the latent space by fitting the respective method
        Create new samples from the latent space $Y'$ by drawing (randomly) from the fitted method
    **for** *each element $Y'_i$ in $Y'$* **do**
        Decode $Y'_i$ by passing it through the decoder $g$
**Output:** New sample $X'$

---

## B   Details on the Vine Copula

In the vine copula autoencoder Tagasovska et al. (2019) use *regular-vine (r-vines)*. A r-vine is built of a sequence of linked trees $T_i = (V_i, E_i)$, with nodes $V_i$ and edges $E_i$ for $i = 1, \ldots, d-1$. A $d-$dimensional vine tree structure $V = (T_1, ..., T_{d-1})$ is a sequence of $T-1$ trees if (see Czado 2019):

1. Each tree $T_j = (N_i, E_i)$ is connected, i.e. for all nodes $a, b \in T_i, i = 1, ..., d-1$, there exists a path $n_1, ..., n_k \subset N_j$ with $a = n_1, b = n_k$.

2. $T_1$ is a tree with node set $N_1 = \{1, ..., d\}$ and edge set $E_1$.

3. For $i \geq 2$, $T_j$ is a tree with node set $N_i = E_{i-1}$ and edge set $E_i$ .

4. For $i = 2, ..., d - 1$ and $\{a, b\} \in E_i$ it must hold that $|a \cap b| = 1$.

An example of a five-dimensional vine tree structure is given below in Figure 7. Note that the structure has to be estimated and multiple structures are possible. For details on vine copula estimation, see Czado (2019); Joe (2014); Bedford & Cooke (2002).

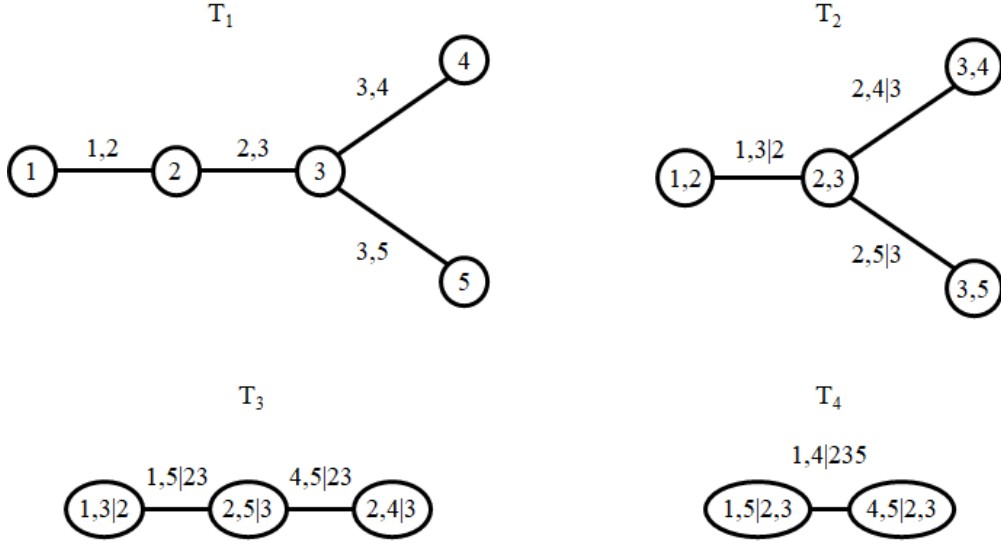

Figure 7: Example of a vine copula tree structure $T_1 - T_4$ for five dimensions.

## C   Asymptotics of the Empirical Beta Copula

Theorem C.1 gives the asymptotic behavior of the empirical beta copula.

**Theorem C.1** (Asymptotics of the empirical beta copula)**.** *Let the copula $C$ have continuous first-order partial derivatives $\dot{C}_j = \delta C(\mathbf{u})/\delta u_j$ for each $j \in \{1, \ldots, d\}$ on the set $I_j = \{\mathbf{u} \in [0, 1]^d : 0 < u_j < 1\}$. The corresponding empirical copula is denoted as $\mathbb{C}_n$, with empirical copula process $\mathbb{G}_n = \sqrt{n}\left(\mathbb{C}_n(\mathbf{u}) - C(\mathbf{u})\right)$ and empirical beta copula $\mathbb{C}_n^\beta$ with empirical beta copula process $\mathbb{G}_n^\beta = \sqrt{n}\left(\mathbb{C}_n^\beta(\mathbf{u}) - C(\mathbf{u})\right)$. Suppose $\mathbb{G}_n \rightsquigarrow \mathbb{G}$ for $n \to \infty$ to $\mathbb{G}$ in $l^\infty([0, 1]^d)$, where $\mathbb{G}$ is a limiting process having continuous trajectories almost surely. Then, in $l^\infty([0, 1]^d)$*

$$\mathbb{G}_n^\beta = \mathbb{G}_n + o_p(1), n \longrightarrow \infty.$$

*Proof.* See Segers et al. 2017 Section 3. □

In short, Theorem C.1 states that the empirical beta copula has the same large-sample distribution as the empirical copula and, thus, converges to the true copula. However, the empirical beta copula performs better for small samples. Segers et al. 2017 demonstrate that the empirical beta copula outperforms the empirical copula both in terms of bias and variance.

## D    Implementation

### D.1    Implementation of the Autoencoder

We implemented the experiments in Python 3.8 Van Rossum & Drake Jr (1995) using `numpy 1.22.0, scipy 1.7.1„ scikit-learn 1.1.0 and pytorch 1.10.1` Harris et al. (2020); Virtanen et al. (2020); Pedregosa et al. (2011); Paszke et al. (2019). The AEs were trained using the Adam optimizer with learning rate 0.001 for MNIST and 0.0005 for SVHN and CelebA. A weight decay of 0.001 was used in all cases. Batch sizes were fixed to 128 (MNIST), 32 (SVHN) and 100 (CelebA) samples for training, while the size of the latent space was set to 10 (MNIST), 20 (SVHN) and 100 (CelebA) according to the data sets size and complexity. Training was executed on a separate train set and evaluated on a hold-out test set of 2000 samples, similar to Tagasovska et al. 2019. For comparison with the VCAE and performance metrics, we have resorted to the implementation from Tagasovska et al. 2019 and Xu et al. 2018. The architectures for all networks are described in Appendix D.2. We trained the autoencoders on an NVIDIA Tesla V100 GPU with 10 Intel Xeon Gold 6248 CPUs. The experiments are executed afterward on a PC with an Intel i7-6600U CPU and 20GB RAM.

### D.2    Architectures of Autoencoders and VAE

We use the same architecture for EBCAE, VCAE, and VAE as described below. All models were trained by minimizing the Binary Cross Entropy loss.

**MNIST**

**Encoder:**

$$
\begin{aligned}
x \in R^{32\times32} \to Conv_{32} &\qquad \to BN \to ReLu \\
\to Conv_{64} &\qquad \to BN \to ReLu \\
\to Conv_{128} &\qquad \to BN \to ReLu \\
&\qquad \to FC_{10}
\end{aligned}
$$

**Decoder:**

$$
\begin{aligned}
y \in R^{10} \to FC_{100} \to ConvT_{128} &\qquad \to BN \to ReLu \\
\to ConvT_{64} &\qquad \to BN \to ReLu \\
\to ConvT_{32} &\qquad \to BN \to ReLu \\
&\qquad \to FC_{1}
\end{aligned}
$$

For all (de)convolutional layers, we used $4 \times 4$ filters, a stride of 2, and a padding of 1. $BN$ denotes batch normalization, $ReLU$ rectified linear units, and $FC$ fully connected layers. Last, $Conv_k$ denotes the convolution with $k$ filters.

**SVHN**

In contrast to the MNIST dataset, images in SVHN are colored. We do not use any preprocessing in this dataset.

**Encoder:**

$$
\begin{aligned}
x \in R^{3\times32\times32} \to Conv_{64} &\qquad \to BN \to ReLu \\
\to Conv_{128} &\qquad \to BN \to ReLu \\
\to Conv_{256} &\qquad \to BN \to ReLu \\
&\qquad \to FC_{100} \to FC_{20}
\end{aligned}
$$

**Decoder:**

$$y \in R^{20} \to FC_{100} \to ConvT_{256} \qquad \to BN \to ReLu$$
$$\to ConvT_{128} \qquad \to BN \to ReLu$$
$$\to ConvT_{64} \qquad \to BN \to ReLu$$
$$\to ConvT_{32} \qquad \to BN \to ReLu$$
$$\to FC_1$$

Notations are the same as described above.

### CelebA

In contrast to the MNIST dataset, images in CelebA are colored. Further, we first took central crops of $140 \times 140$ and resize the images to a resolution $64 \times 64$.

**Encoder:**

$$x \in R^{3 \times 64 \times 64} \to Conv_{64} \qquad \to BN \to LeakyReLu$$
$$\to Conv_{128} \qquad \to BN \to LeakyReLu$$
$$\to Conv_{256} \qquad \to BN \to LeakyReLu$$
$$\to Conv_{512} \qquad \to BN \to LeakyReLu$$
$$\to FC_{100} \to FC_{100}$$

**Decoder:**

$$y \in R^{100} \to FC_{100} \to Conv_{512} \qquad \to BN \to ReLu$$
$$\to ConvT_{256} \qquad \to BN \to ReLu$$
$$\to ConvT_{128} \qquad \to BN \to ReLu$$
$$\to ConvT_{64} \qquad \to BN \to ReLu$$
$$\to ConvT_{32} \qquad \to BN \to ReLu$$
$$\to FC_1$$

*LeakyReLU* uses a negative slope of 0.2, and padding was set to 0 for the last convolutional layer of the encoder and the first of the decoder. All other notations are the same as described above.

### D.3  Implementation of Real NVP

In our study, we used a Real NVP (see Dinh et al. 2017) to model the latent space of the autoencoder and serve as a benchmark. For all data sets, we use spatial checkerboard masking, where the mask has a value of 1 if the sum of coordinates is odd, and 0 otherwise. For the MNIST data set, we use 4 coupling layers with 2 hidden layers each and 256 features per hidden layer. Similarly, for the SVHN data set, we also use four coupling layers with two hidden layers each and 256 hidden layer features. Lastly, for the CelebA data set, we use four coupling layers with two hidden layers each and 1024 hidden layer features. For all data sets, we applied a learning rate of 0.0001 and learn for 2000 epochs.

## E  Image Interpolation of the Autoencoder

We show that our used autoencoder learned a relevant and smooth representation of the data by interpolation in the latent space and, thus, modeling the latent space for generating new images is reasonable. For example, consider two images A and B with latent variables $y_{A,1}, ..., x_{A,100}$ and $y_{B,1}, ..., y_{B,100}$. We now interpolate linearly in each dimension between these two values and feed the resulting interpolation to the decoder to get

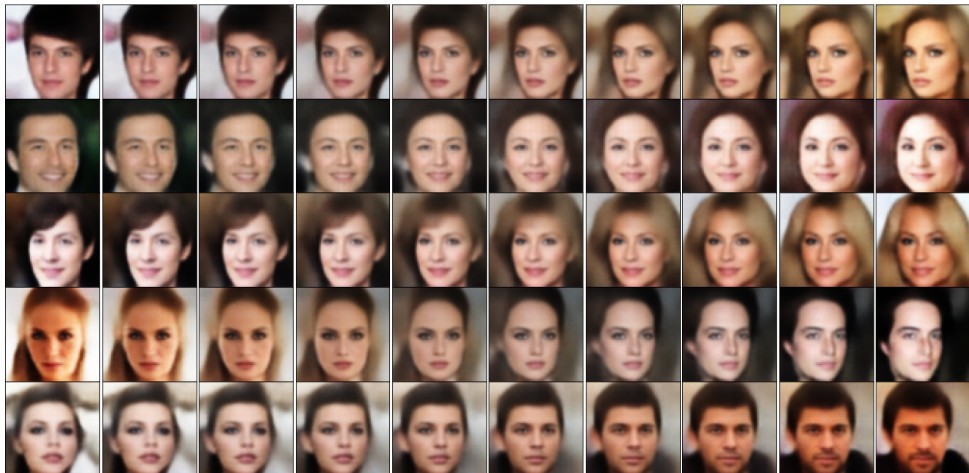

Figure 8: Interpolation in the latent space of samples of the autoencoder.

the interpolated images. Each row in Figure 8 shows a clear linear progression in ten steps from the first face on the left to the final face on the right. For example, in the last row, we see a female with blonde hair slowly transforming into a male with a beard. The transition is smooth, and no sharp changes or random images occur in-between.

## F   Additional Experiments

### F.1   Numerical Assessment of Methods on CelebA

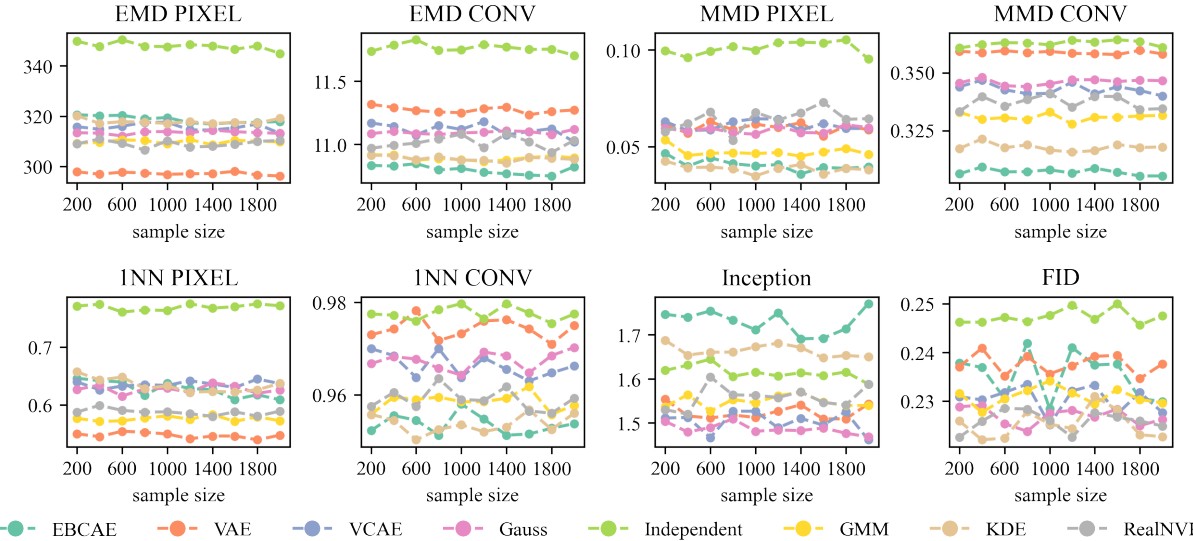

Figure 9: Performance metrics of generative models on **CelebA**, reported over latent space sample size. Note that they only differ in the latent space sampling and share the same autoencoder.

### F.2 Numerical Assessment of Methods on MNIST

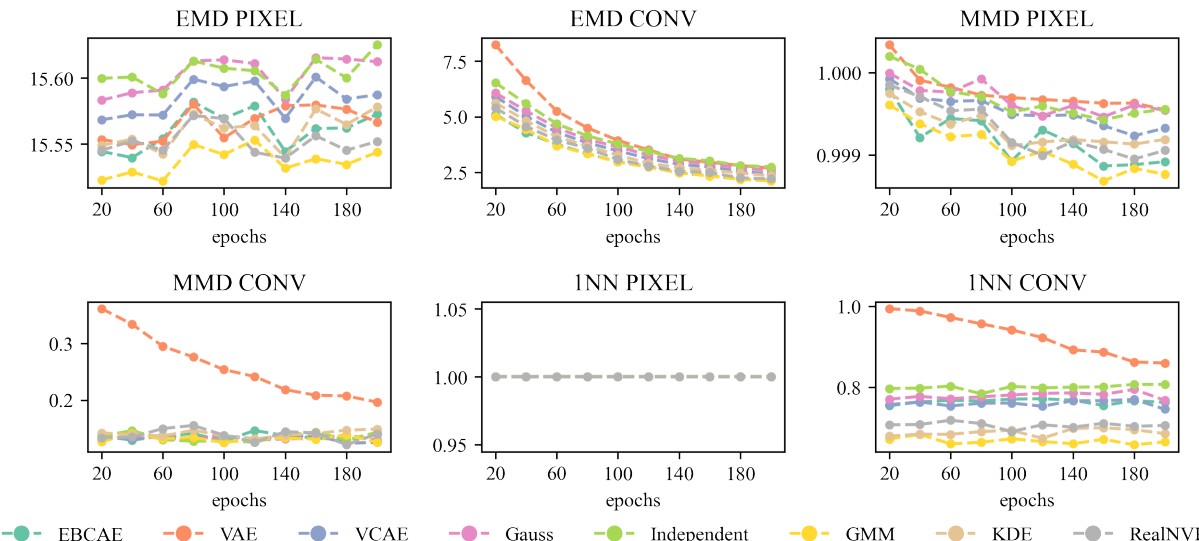

Figure 10: Performance metrics of generative models on **MNIST**, reported over epochs computed from 2000 random samples. Note that they only differ in the latent space sampling and share the same autoencoder.

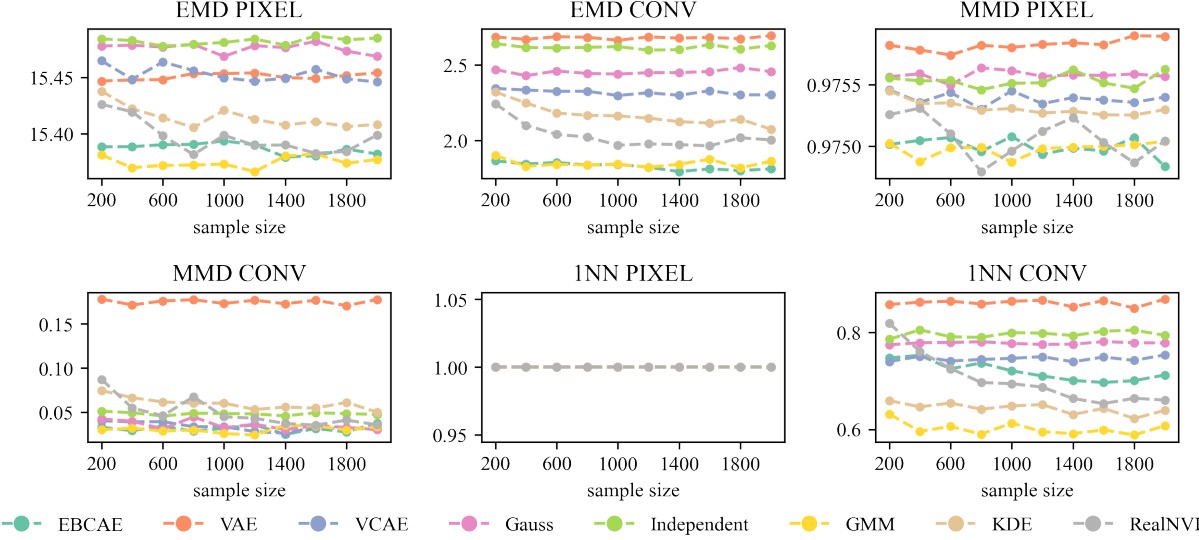

Figure 11: Performance metrics of generative models on **MNIST**, reported over latent space sample size. Note that they only differ in the latent space sampling and share the same autoencoder.

## F.3    Numerical Assessment of Methods on SVHN

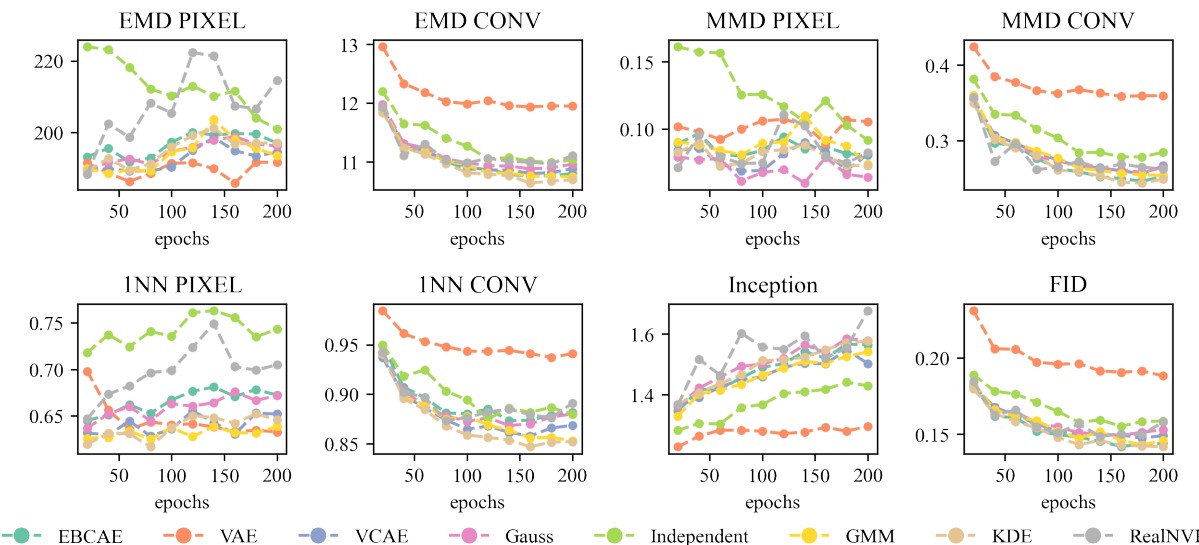

Figure 12: Performance metrics of generative models on **SVHN**, reported over epochs computed from 2000 random samples. Note that they only differ in the latent space sampling and share the same autoencoder.

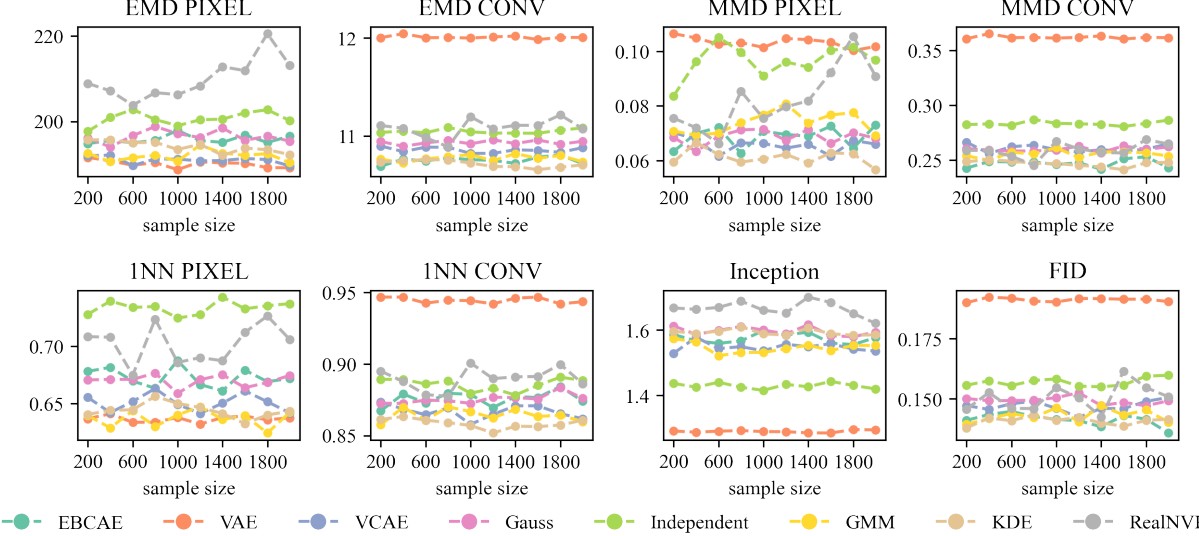

Figure 13: Performance metrics of generative models on **SVHN**, reported over latent space sample size. Note that they only differ in the latent space sampling and share the same autoencoder.

## F.4    Generated Images from SVHN

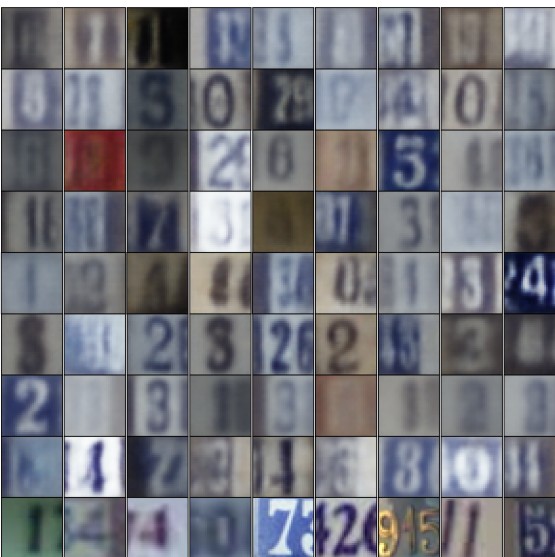

Figure 14: Comparison of synthetic samples of different Autoencoder models. $\mathbf{1}^{st}$ **row:** Fitted normal distribution, $\mathbf{2}^{nd}$ **row:** Independent margins, $\mathbf{3}^{rd}$ **row:** KDE-AE, $\mathbf{4}^{th}$ **row:** GMM, $\mathbf{5}^{th}$ **row:** VCAE, $\mathbf{6}^{th}$ **row:** EBCAE, $\mathbf{7}^{th}$ **row:** VAE, $\mathbf{8}^{th}$ **row:** Real NVP, **Last row:** original pictures.

# G    Code

Code will be provided here. [link to the repository will be inserted]

