# OpenReview forum: "Sampling from the latent space in Autoencoders: A simple way towards generative models?"
_TMLR — Rejected by TMLR_

### Review · Reviewer_q21U · 2023-04-22

**Summary Of Contributions:**

While auto-encoders learn high-quality data representations, they are not generative models. Tagasovska et al., 2019 proposed a method to convert a trained auto-encoder into a generative model using a three-step process:
Learn an autoencoder.
Use training data representations as observed samples and learn a distribution in the representation space.
Sample from the learned representation distribution and decode to sample novel data points.

In this paper, the authors examine different existing methods of learning the representation distribution. The paper aims to understand the different properties of existing methods and analyze the trade-offs. Additionally, they propose a new method, the Empirical Beta Copula Autoencoder (EBCAE). Overall, the paper aims to develop a clearer understanding of how different methods fare against each other and the advantages and trade-offs of using one over the other.

**Audience:**

No

**Broader Impact Concerns:**

Not applicable.

**Claims And Evidence:**

No

**Requested Changes:**

See the weakness section.

**Strengths And Weaknesses:**

# Strengths

- I like the overall direction of this work. We need research that attempts to make sense of several possible methods for a particular problem. Such analyses are essential in general and fit perfectly in the scope of TMLR.

# Weakness

* I found the paper could have been written better. The manuscript will benefit immensely from a thorough rewrite. For instance
     * "We argue that imposing restrictions on the distribution should be avoided and that more flexible approaches for modeling the latent space seem beneficial." I need help figuring out what the argument is here. Why should this be better?
     * The authors use latent space and representation interchangeably. Nothing is latent (or "hidden") about the representations of the autoencoders. The mapping from the data point to the representation is deterministic. I suggest that authors clarify their notation a bit more and set up the context of the paper early.
     * "conditioning set $D_e$ and conditional probabilities, e.g.,.." There needs to be an explanation of the conditioning set. In general, several things were hard to understand about the copula background.
     - "For each copula, encoding the dependence of two conditional variables, any bivariate copula model, including non-parametric modeling approaches (as done by Tagasovska et al. 2019) can be chosen." Complex sentences like this can make it harder to follow through with the author's argument.

- It felt the authors were very defensive about not proposing a new method, which is perfectly fine. Instead, using the landscape in the first few pages is vital to bring out the key points from the analysis. No section clearly articulates the contributions of the paper. What were the specific insights that the authors found in their extensive experiments? One has to dig deep into the paper's experiments section to understand the key takeaways.

- I found Figure 2 extremely hard to parse. There are nine methods. You must match the method to the row whenever you want to compare anything. The core contribution of this work is the analysis. Therefore, the analysis should be easier to parse.

- Authors conduct different experiments; however,  I found them incomprehensible sometimes. I suggest restructuring the experimental section such that different settings are brought out better. For instance, the setup related to Figure 4 and Figure 5 is hidden in the dense text.

---

> ### Author Response · Authors · 2023-05-11
> **Answer to the Reviewer**
>
> We again thank the Reviewer for the evaluation of the paper and answer the questions/ comment on the weaknesses below. Please also see the General Answer to the Reviewers above.
>
>  Weaknesses:
> 1. We thank the reviewer for sharing their impression.  We improved the paper by adding additional explanations and restructuring the experimental section of the paper. We think the paper is now much clearer. Furthermore, we also will incorporate the suggestions of the Reviewer in a revised version.
> 2. We are now more clear about the contribution and intention of the paper in the introduction. We now further added a short, more comprehensive summary at the end of the introduction.
> 3. We thank the reviewer for their impression. We now added the method's name directly to the corresponding line in the figure. This highly improves the speed of perception of the figure and makes it easier to parse for the reader.
> 4. We described the setup related to the figures in the caption more detailed and give the experimental section a clearer structure by including additional paragraphs and more explanatory text.

---

### Review · Reviewer_nEGu · 2023-05-03

**Summary Of Contributions:**

This paper studies the behaviour of autoencoder with diverse latent space modeling methodologies such as multivariate Gaussian distribution, Gaussian mixture model, kernel density estimates, copula-based methods and the RealNVP. The authors argue that turning traditional autoencoder with latent space modeling can be alternative of generative models. The authors conducted experiments on MNIST, CelebA, and SVHN datasets to verify their claim.

**Audience:**

No

**Claims And Evidence:**

Yes

**Requested Changes:**

- The methodology part in Section 3 should be more specified. What is the protocol of latent space modeling with autoencoders? It would be better if there are additional pseudo-codes for the overall learning and sampling processes.

- The experiment should be conducted stand-alone generative model vs AE + generative model. For example, GMM in the data space vs GMM in the latent space of AE.


**Strengths And Weaknesses:**

Strengths:

- The authors investigated various generative model-based latent space modeling approach in the autoencoder.

- If I understand correctly, the methodology that the authors utilized in the experiment is straightforward.

- The authors provide appropriate evidences for their claim.


Weaknesses:

- It seems the authors' arguing point is not much different from modeling data in the reduced dimension space of dimentionality reduction algorithms. In other words, instead of direct modeling with GMM, the considered methodology is that (1) first, reduce dimension with autoencoder, (2) modeling the reduced dimension space with GMM, and (3) revert back to the original data space.

- The paper is written hard to understand the message.


Questions:

- What are "Independent" and "Independent margins" methodologies stands for? Are they "Independent modeling approach" discussed in the KDE section?

- What is the advantage of the empirical beta coplua that the authors proposed (but not argued as the contribution in this work)?

- All the experiments are conducted in the settings of AE + latent space modeling, except the VAE and RealNVP, right? Isn't the same approach can be utilized with RealNVP, i.e., AE + RealNVP?

.

---

> ### Author Response · Authors · 2023-05-11
> **Answer to the Reviewer**
>
> We again thank the Reviewer and answer the questions below. Please also see the General Answer to the Reviewers above.
>
> Questions:
> 1. Yes, they are the "independent modeling approach" discussed in the KDE section. We clarify this in the revised version of the paper.
>
> 2. In contrast to the vine copula approach, the empirical beta copula models the full, untruncated distribution of the data at hand. It further is non-parametric and straightforward to sample from. Last, it allows the recombination of different marginal distributions and dependence structures in the latent space.
>
> 3. Yes. In fact, this is what we did. The RealNVP is used in the latent space to be comparable to the other methods.
>
> Requested Changes:
> 1. We added additional pseudo code for the overall approach in the appendix.
> 2. The data space is 64x64x3=12288 dimensions (in the case of the CalebA), which makes the application of the proposed methods not feasible. Thus, we do not present the stand-alone results of the generative models. However, we believe the results are comparable since all methods share the same autoencoder.

---

### Review · Reviewer_kZsC · 2023-05-06

**Summary Of Contributions:**

This paper considers the problems of creating generative models by learning to sample from the latent space of auto-encoders. They also propose a method called the Empirical Beta Copula Autoencoder which uses an empirical beta copula for modeling and sampling from the latent space of an auto-encoder. The paper conducts a few experiments to compare several AE-based samplers over several measures.

**Audience:**

Yes

**Claims And Evidence:**

No

**Requested Changes:**

1. The authors should first provide a solid theoretical grounding to the framework of turning an AE into a sampler. There are several papers that do this. For instance, authors can refer to https://arxiv.org/abs/1711.01558, https://proceedings.mlr.press/v161/mondal21a.html, or https://arxiv.org/abs/1903.05789. Without these theoretical underpinnings, placing the present work in the proper context is difficult.

2. A lot of important baselines are missing both in section 2 and experimental comparisons- https://arxiv.org/abs/1705.07120, https://arxiv.org/abs/1903.12436.

3. Authors mention - "Vine copulas offer a solution to this problem and decompose the multivariate
density as a cascade of bivariate building blocks organized in a hierarchical structure" - The context does not describe the "problem" that the Authors are referring to.

4. The text suddenly jumps to the section on Empirical Beta Copula Autoencoder without motivating the need for it.

5. There is no justification (theoretical) to claim the advantage of the considered distribution on the latent space.

6. The datasets considered for experiments are very toyish. Some experiments on large-scale corpus such as Imagenet have to be demonstrated.

7. In Fig. 3, the range of FID seems unreasonable.

8. Does the proposed model induce interpretable semantic directions on the latent space?

9. One important question in the context of AE as a generative model is the following - Should one induce a sampler on the latent space jointly with the AE training or it has to be done post-hoc like the one presented in the paper? Studies have shown that the former approach is more principled and yields better results. Did the Authors try such approaches?





**Strengths And Weaknesses:**

Strengths:

1. Considers a critical problem in the AE community.

2. Well written and easy to comprehend.

Weaknesses:

1. This paper does not offer any heretofore unknown information. Most of the observations made in this paper are very well-known to an expert in the community.

2. It provides no theoretical insights into the proposed new model.

3. Comparisons and experiments are rather weak.

---

> ### Author Response · Authors · 2023-05-11
> **Comments on requested Changes**
>
> We again thank the Reviewer and answer the requested changes below. Please also see the General Answer to the Reviewers above.
>
> Requested Changes:
>
> 1. We thank the reviewer for the two additional references regarding WAEs we have not been mentioning so far. We now include these papers in our work.
>
> 2. We thank the reviewer for the references. However, the mentioned baselines do either not use a plain AE -- hence are not a fair/natural benchmark from our viewpoint -- or are based on a Gaussian Mixture Model (GMM), which we already included in Section 2 and the experiments while providing the mentioned reference.
>
> 3. The problem to which we refer is that only a few multivariate copula models exist, hence modeling data is difficult. We now clarify this in the previous sentence.
>
> 4. We thank the reviewer for sharing his/her impression at the start of this section. We now give more context and motivate the usage of the EBC.
>
> 5. Thank you for pointing this out. To our knowledge, there is no theoretical analysis of the distribution in the latent space available. We want to point out, that your critique could be made, e.g., also for VAEs which assume a normal distribution in the latent space since the reparametrization trick allows for stable learning. The investigated methods are quite general parametric and non-parametric methods to model (high-dimensional) distributions. Of course, if we knew the true distribution from theoretical analysis, we could use that straight away.
>
> 6. We agree that the used datasets are not the most challenging and complex datasets. However, they incorporate different characteristics, levels of complexity and latent dimensions (10,20,100).
>
> 7. We thank the reviewer for this comment and excuse the inaccuracy in our explanation. In our evaluation, we follow the framework Xu et al. (see https://arxiv.org/abs/1806.07755) and calculate the FID over ResNet-34 softmax probabilities, which results in the shown range of the FID scores. However, we will also calculate the 'standard' FID values and add them to a revised version of the paper.
>
> 8. Thank you for asking this. We are not sure whether this is possible for all methods investigated. The observed behavior for splitting attributes between copula and marginals seems to be a first indication that this might be possible for copula-based methods. However, this is subject to future research.
>
> 9. We are aware of the discussion around this issue and that studies have shown better results for joint training. However, we focused on the post-hoc approach since the training of the autoencoder is untouched and it is more straightforward (as it was the intention of our study). Additionally, for non-parametric methods, joint training seems to be infeasible.

---

### Review · Reviewer_7FWj · 2023-05-06

**Summary Of Contributions:**

This paper empirically compare different post-hoc methodologies to model the latent space of an autoencoder in order to turn it into a generative model. The empirical beta copula is suggested as an efficient and flexible alternative. These two-steps generative models are also benchmarked against a VAE and a RealNVP. Results on three datasets (MNIST, CelebA, and SVHN) illustrate that the two-steps approach is viable.


**Audience:**

Yes

**Broader Impact Concerns:**

NA.

**Claims And Evidence:**

No

**Requested Changes:**

* Not enough details is provided on the estimation techniques and fitted model parameters.
* A more thorough analysis of the *built-in* conditional sampling feature for EBCAE.
* Clarify how the images attributes are split between the marginals and the copula for the copulas models is interesting. Is this anecdotal?
* Clarify the theoretical suggestion in the last section on the structural link between copulas and NF.

**Strengths And Weaknesses:**

Strengths:
* An easy to read and follow comparative empirical analysis.
* An efficient methodology, the EBCAE, is proposed.

Weaknesses:
* Not a new methodology, only the EBC appears to be novel in this context.
* No new experiments, standard and small datasets. No SOTA results.
* No theoretical contribution.

Questions/Suggestions:
* The EBCAE appears able to produce more realistic samples but with less variability. Vice versa for some others methodologies. What is more important?
* It is not clear why the fitted vine-copula performs relatively poorly and that its spanned latent space (Figure 5) is so different from the original data. Could it be an estimation and/or a model-selection issue?

---

> ### Author Response · Authors · 2023-05-11
> **Answer to Questions and Weaknesses**
>
> We again thank the Reviewer and answer the questions below. Please also see the General Answer to the Reviewers above.
>
> Weaknesses:
>
> We agree that using a plain vanilla autoencoder does not compare to actual SOTA models and results in generative learning. However, we think the paper is highly interesting for the majority of TMLR's audience, since it may help to understand the pros and cons of various methods. We further want to emphasize that the approaches could be transferred to other, more sophisticated, state-of-the-art autoencoders, which may be beneficial to avoid too strong regularization schemes to fit a given prior. However, this is subject to future research.
>
>
> Questions:
>
> 1. From our perspective, the trade-off between variability and realistic samples depends on the individual application and no one-fits-all answer is suitable. Using the marginals and dependence structure from different sample categories in the EBCAE as pointed out in the paper could offer a promising solution to this dilemma.
>
> 2. In general, both issues could cause a non-optimal fit of a vine copula. However, in our case, the non-optimal fit is a model selection issue, since non-parametric copulas were used for all pairwise copulas of the vine.  Further, the vine structure is truncated as in the original paper (see https://proceedings.neurips.cc/paper_files/paper/2019/hash/15e122e839dfdaa7ce969536f94aecf6-Abstract.html), which may be the main driver of this behavior. The observation of the reviewer, therefore, is a general problem of the vine-copula approach.
>
>
> Requested Changes:
>
> 1. Note that we mainly use non-parametric methods, which do not have model parameters. In the case of the Gaussian Distribution and GMM, the parameters are $100\times 100$ covariance matrices which are hard to parse and therefore not shown. However, the code is openly available.
>
> 2. The targeted sampling works by  randomly choosing among rows in the rank matrix of original input samples that belong to one specific sub-group. Therefore new samples are only generated from the input samples from one subgroup. We clarify that in the paper now.
>
> 3. Thank you for asking this. We thought it is interesting to study the influence of changing marginals and dependence structures when using copulas. Therefore, we reported the results. However, we think this split might be individual for each autoencoder and latent space under investigation. In summary, we are not yet sure whether this is anecdotal or a general scheme and, thus, leave this for further research.
>
> 4. We thank the reviewer for spotting this. This is an old comment, we forgot to erase. We do not think anymore, that there is an easy structural link of copulas and NFs.

---

### Author Response · Authors · 2023-05-11
**General Answer to the Reviewers**

We thank the reviewers and the AE for the thoughtful evaluations of our paper. The insightful comments and constructive feedback provided have been instrumental in improving the quality and clarity of our work.

 We again want to emphasize that our paper aims to explore the most straightforward approach to transforming an autoencoder (AE) into a generative model. We intentionally did not consider other, more complex methodologies since most of them are fundamentally different models, much more complex, and may not be trained post-hoc after fitting the autoencoder. In contrast to that, our goal is to explore an alternative path towards developing more complex generative models while striving to identify the simplest yet effective model. We also wanted to find out, whether there are promising alternatives to the usage of GMMs in other SOTA models as done, e.g., by https://arxiv.org/abs/1903.12436. While doing so, we came up with the EBCAE as a copula-based method and discovered the trade-off between reproduction and generating new images, which we also analyze for all methods in the paper.

 We answered all the questions of the reviewers below and comment on some issues and requested changes.

---

### Decision · Action_Editors · 2023-06-07

**Recommendation:** Reject

**Comment:**

The paper has gone a through a thorough review and discussion phase. All reviewers and undersigned agree that the paper should be rejected.

The topic is of general interest, the authors have well-written and scoped article so if the authors feel that they can fully address the reviewers' and undersigned comments, then a new submission building on this paper is welcomed.

The authors are encouraged to work on scoping the contributions they want to convey, for example the copula-based method seems to be the most novel contribution here. This could be extended upon in the current setting or used to in a different setting to make an entire new contribution. This is up to you to decide if you want to submit again.

Your AE

**Audience:**

No.

**Claims And Evidence:**

No.

**Resubmission Of Major Revision:**

The authors may consider submitting a major revision at a later time.